# Design and Implementation of a Workshop for Evaluation of the Role of Power in Shaping and Solving Challenges in a Smart Foodshed

Ayaz Hyder [1,*] , Angela Blatt [2] , Allan D. Hollander [3] , Casey Hoy [4,5] , Patrick R. Huber [3] , Matthew C. Lange [6] , James F. Quinn [7] , Courtney M. Riggle [3] , Ruth Sloan [8] and Thomas P. Tomich [3]

1 College of Public Health and Translational Data Analytics Institute, Ohio State University, Columbus, OH 43210, USA
2 Initiative for Food and AgriCultural Transformation, Ohio State University, Columbus, OH 43210, USA; angelacblatt614@gmail.com
3 Food Systems Lab, Department of Environmental Science & Policy, University of California, Davis, CA 95616, USA; adhollander@ucdavis.edu (A.D.H.); prhuber@ucdavis.edu (P.R.H.); cmriggle@ucdavis.edu (C.M.R.); tptomich@ucdavis.edu (T.P.T.)
4 Agroecosystems Management Program, Department of Entomology, Initiative for Food and AgriCultural Transformation, Ohio State University, Wooster, OH 44691, USA; hoy.1@osu.edu
5 Agroecosystems Management Program, Department of Entomology, Initiative for Food and AgriCultural Transformation, Ohio State University, Columbus, OH 43210, USA
6 International Center for Food Ontology Operability Data and Semantics (IC-FOODS), Davis, CA 95616, USA; matthew@ic-foods.org
7 Department of Environmental Science & Policy, University of California, Davis, CA 95616, USA; jfquinn@ucdavis.edu
8 College of Public Health, Ohio State University, Columbus, OH 43210, USA; sloan.930@buckeyemail.osu.edu
* Correspondence: hyder.22@osu.edu; Tel.: +1-614-247-4936

**Abstract:** Current studies on data sharing via data commons or shared vocabularies using ontologies mainly focus on developing the infrastructure for data sharing yet little attention has been paid to the role of power in data sharing among food system stakeholders. Stakeholders within food systems have different interpretations of the types and magnitudes of their own and other's level of power to solve food system challenges. Politically neutral, yet scientifically/socioeconomically accurate power classification systems are yet to be developed, and must be capable of enumerating and characterizing what power means to each stakeholder, existing power dynamics within the food system, as well as alternative forms of power not currently utilized to their full capacity. This study describes the design and implementation of a workshop, which used methods from community-based participatory modeling, to examine the role of power relative to data sharing and equitable health outcomes. Workshop participants co-created several boundary objects that described the power relationships among food system stakeholders and the changes needed to current power relationships. Our results highlight current imbalances in power relationships among food system stakeholders. The information we collected on specific relationships among broad categories of stakeholders highlighted needs for initiatives and activities to increase the types and varieties of power especially across consumers, farmers, and labor stakeholder groups. Furthermore, by utilizing this workshop methodology, food system stakeholders may be able to envision new power relationships and bring about a fundamental re-orienting of current power relationships capable of valorizing food system sustainability/resiliency, especially the health of its workers and consumers.

**Keywords:** power relationships; food system; foodshed; community engagement; ontology; public health

## 1. Introduction

The term *foodshed* [1], like the term watershed, is used to describe a geographic area, in this case, one within which food flows from sources to consumers. Foodsheds have been described at scales ranging from local to global. Concerning the foodshed, the use of the term *power* is varied but does follow some common threads. A review of the existing and relevant literature from public health and food systems research showed that having access to food was historically an indication of power [2]. Over time, power became associated with the ability to control the production, processing, and movement of foods and the resources required for their making. Other writings use *power* to refer more pointedly to an individual's or organization's ability to influence policy change and provide control over some aspect of the food system [3–10]. This influence may come from control over the built environment [6], having an economic advantage [8], and/or the ability of those with influence to maintain the standing they have in a situation [9]. Ostrom defined power as a function of the value of the opportunity and the extent of the control [11]. The term *agency*, which has a long history in institutional and development economics (viz, Cheung's framing of sharecropping contracts as principal-agent problems [12]), connotes an individual's ability to advocate [4] for themselves and to act as a semi-autonomous agent within social-constructed systems of economic and political constraints.

Different disciplines have emphasized different, yet interrelated, aspects of the role of power in the food system. The role of differences in the level of power between small farmers and large corporations [4,8] or between farm laborers and landowners within the foodshed has been a central focus of agrarian studies, particularly about food sovereignty. For instance, in Argentina, there are many farm laborers whose livelihoods depend on landowners' willingness to hire and pay them [13]. This is not an uncommon phenomenon; it is also an issue for migrant workers in North America [14], where employers exert power over undocumented workers, leveraging the threat of deportation. Similarly, power and related conflicts are common themes in the literature on natural resource management and international agricultural development across the Global South [15,16]. In social work, power relations are typically described on a local or individual scale in terms of oppression and empowerment of the oppressed [13]. From a political economy perspective, the role of power during policy formulation and implementation [2] is a central concern. Take, for instance, a large corporation or company's opposition to policy changes that may force them to change product formulations to be more nutritious but, instead, the corporation leverages its power to deploy the "part of the solution" strategy. In the "part of the solution" strategy, companies (e.g., Mars, Unilever, Kellogg's, and Coca-Cola) have pushed back to acquire supposedly healthy brands and made commitments to reformulate their products to reduce sugar or sodium in their products [17]. Similarly, several wealthier companies control consumer choice architecture and can use their financial power to exert political power, allowing them to influence policy, often to block change [8,18]. In political science and ecology (with the perspective of looking at ecology and society, rather than biotic aspects of the environment), interdependencies within systems (e.g., political, administrative, and social-ecological) occur due to power differences [19]. A common thread among these different perspectives on power from different fields of study is that power mediates changes in the system including the type, speed, timing, and scale of the changes, which type of changes occur or do not occur, how policy changes are implemented and by whom, and who benefits and who loses.

Power imbalances are ubiquitous across the food system, crossing local, regional, national, and global scales. These imbalances have consequences, often leading to adverse social, economic, and health outcomes, political marginalization, impoverishment, and exploitation of those with less power [2], and reinforcing inequalities and injustices, leading to social tension and, sometimes, civil unrest [4,13]. In the case of *Tareferos*, a group of farmworkers in Argentina, exploitation by farm owners meant the laborers had limited access to healthcare or retirement funds. These problems are common for farm workers and small-scale farmers around the world. For farmers in Honduras, loss of land meant

loss of food sovereignty and food insecurity. Power imbalances and policy choices often are mutually reinforcing. In the United States, "redlining", which is a structurally racist policy meant to keep Black people and other racial/ethnic minorities out of neighborhoods where White people live, have perpetuated social, political, and economic inequalities and played a role in the development and spatial arrangement of "food deserts" and other areas with reduced food security [6]. A study undertaken in Australia found that power imbalances in favor of supermarkets negatively affected the nutritional value of the food available to consumers, impacting public health outcomes [7]. In short, power imbalances in food systems (and other systems, too) create a variety of complex, far-reaching problems, many of which can be difficult to identify, understand, and address. Failure to address these manifestations of power imbalances perpetuates social justice and arguably undermines food system sustainability and resilience [20].

Redistribution of power is always challenging, if not impossible. In public health, power imbalances have been identified as a barrier to deliberations about the community health [21], which makes it harder to even start the conversation about redistribution of power within a community. A prerequisite to addressing these challenges within the food system is the engagement of diverse voices and the participation of marginalized stakeholders. Community engagement is important for addressing power imbalances, because it challenges the structural powers that oppress marginalized stakeholders, and shifts the decision-making authority from those with power to those with less or no power [22]. Multiple factors need to be considered to create opportunities (sometimes called "safe spaces") where it is possible to truly hear diverse voices and engage in collective efforts aimed toward positive change. Creating such opportunities in a foodshed is complicated by several factors. One factor is that foodsheds are complex. There are many stakeholders within the foodshed, including producers (e.g., farmers), processors (e.g., butchers), distributors (e.g., transportation), retailers (e.g., retail food stores), consumers (e.g., households) and resource management (e.g., food waste recovery). Each of these stakeholders is affected directly by challenges in the foodshed and could benefit (or suffer) from changes in the food system. Another factor is the relationship between equity and sustainability and how this relationship affects each stakeholder and interactions between stakeholders in the foodshed. Somewhat paradoxically, the pursuit of a more equitable food system may involve tradeoffs with sustainability, and vice-versa [23]. Additionally, consumers and producers with higher income and more political and economic connections (social capital) have greater power than those with lower income and fewer connections [20]. Those with power typically are the voices that dominate discourses around the food system and health policy [24,25]. As a result of these factors, it is difficult to amplify the voices of those with less power and include their views in an authentic manner. Empowering the powerless in the food system by including them in the design, governance, operation, and policies that shapes foodsheds, including both the structure and function of food systems, may lead to a more sustainable and sustainable foodshed.

Understanding what power means to each stakeholder in a foodshed is a key step towards identifying existing power relationships. Identifying existing power relationships and reimagining those relationships may help to achieve a more equitable and resilient food system. Of course, it is the exercise of power in the pursuit of narrow self-interest and in opposition to broader common interests (say in public health, childhood nutrition, environmental health) that is the canonical problem underpinning so many policy failures. Through the development of methods and tools for understanding these relationships and convening and lifting marginalized voices, the long-term goal is to mobilize food system stakeholders for the identification and development of collaborative solutions that shift outcomes toward the broader public interest [20,24,25].

**One tool for bringing together diverse voices is the use of community-based system dynamics, specifically group model building**. The community-based system dynamics approach was developed as part of system dynamics modeling, which is a methodology from the field of systems science [26–32]. In the community-based system dynamics ap-

proach, the dynamics, feedbacks, and nonlinearities within a system are identified by community participants using facilitated discussions and group activities [33]. Bringing together these diverse voices of participants allows participants to develop solutions to problems within a system that they care about and work together in, but may not know how each organization's decisions or policies affect other organizations and, eventually, the entire system [33]. This approach to problem-solving prioritizes inclusion and collaboration with multiple stakeholders, with the goal of co-creating solutions to challenges that benefit multiple stakeholders. These solutions ideally transcend the power differences concentrated in the vested interests of each organization or stakeholder by finding a common good. Perhaps, more realistically, and in the context of a foodshed, the hope is to mitigate the impacts of power imbalances by engaging diverse voices and mobilizing marginalized communities to address food system challenges that injure them directly.

Examples of the community-based system dynamics approach, in particular, group model building, in the food system literature include studies on food insecurity [34], food consumption [35], obesity [32,36], food supply and demand [37], food-related policymaking [38], and program evaluation [39]. In each of these studies, input for the modeling was gathered from a variety of stakeholders through facilitated discussions. The facilitated discussions generated various types of boundary objects. In some cases when data were used to build a system dynamics model, the authors used those boundary objects and translated model-driven insights into policy actions. A boundary object has many definitions [40], but we are using it here to mean textual and graphical representations of ideas, perspectives, and mental models of group model building workshop participants [29]. Boundary objects may serve to identify new ways to build data discovery and data sharing capabilities among food system stakeholders.

The community-based system dynamics approach offers a way to develop boundary objects in a manner that mitigates power imbalances and, in the long-term, may facilitate data discovery and data sharing among stakeholders. Food system stakeholders often do not know about each other's data holdings, which prevents them from asking new questions or sharing their data with others. If we assume that all food system stakeholders have some power to share data, then it is reasonable to also assume that the level of power varies and remains unknown to other food system stakeholders. Based on Ostrom's definition of power, the value of the opportunity (i.e., sharing data) and the extent of control (i.e., the capacity to share data) is, perhaps, most appropriate to bring up again because food system stakeholders may vary in how they value data sharing. Similarly, the extent of control to share data may vary due to different capacities and access to resources needed to share data. Therefore, identifying power relationships among data holders and re-orienting those relationships might be one way to enable more data sharing among food system stakeholders to solve foodshed challenges. Identifying such relationships using a methodology that minimizes power imbalances among stakeholders is a novel feature of the community-based system dynamics approach.

While current studies on data sharing via data commons or shared vocabularies using ontologies (e.g., FoodON [41]) mainly focus on developing the infrastructure for data sharing, little attention has been paid to the role of power in data sharing among food system stakeholders. To make progress in addressing this problem, we set out to conduct a group model building workshop with food system stakeholders to start to develop a class called 'Power' within an existing ontology, known as PPOD for "People, Project, Organizations, and Datasets" [1,42,43]. An *ontology* is a set of concepts within a specific area or domain, which describes the properties of those concepts and the relationships between the concepts. A *class* is a core element comprising an ontology [44]. The PPOD ontology was developed for foodshed and food systems research and applications. Ultimately, we aim to integrate the class Power within the PPOD ontology. A novel feature of our study is extending the community based system dynamics approach, specifically group model building scripts, for enhancing the PPOD ontology. This is a novel feature, because typically

ontology development is an academic exercise where community stakeholders are not directly involved in the process [45].

The addition of the class Power is a foundational step in eliminating barriers to including diverse voices in shaping and solving challenges in the foodshed through data sharing, transparency, discovery, data-informed debate, and co-creation of data-enabled solutions. It is conceivable that such a classification system could enable the re-imagining of existing power relationships within the food systems to make progress towards greater equity, diversity, transparency, and accountability of food system actors. It is hoped that our work can provide tools to enhance organizing and collective action to confront powerful vested interests standing in opposition to the public good.

## 2. Materials and Methods

This study describes the design and implementation of a workshop to evaluate the role of power as it relates to data sharing and equitable health outcomes. The workshop is based on the community based system dynamics approach. Along with discussing how we have started to conceptually develop the class, Power, using boundary objects that emerged from the workshop, we also discuss how it may be used to re-orient the foodshed toward a more equitable food system and equitable health outcomes.

### 2.1. Setting

We organized a workshop at the second annual IC-FOODS conference held in Davis, California on 22–25 March 2019. IC-FOODS is an organization that "brings together the brightest minds in ontological, computational, and mathematical modeling from around the world, together with domain experts whose work resides along the Environment → Agriculture → Food → Diet → Health knowledge spectrum". [46] The conference focused on bringing together local, regional, national, and international stakeholders from multiple disciplines, such as computer science, medicine, public health, agriculture, public policy, and sectors, such as business and industry, non-profits, and government. The in-person workshop, entitled "Designing and assembling the framework of ontologies for tracking food through the food system" was held on the last day of the conference.

### 2.2. Workshop Participants

The workshop participants included individuals who attended the IC-FOODS conference and had signed up to attend the workshop after the conference. Broadly, the participants represented diverse backgrounds including community-based organizations, food system advocates, academia/researchers (computer scientists, informatics, data scientist, agriculture, nutrition, food systems), industry (technology companies, startups) professional associations, government agencies, and food system policymakers/analysts. Some participants represented multiple backgrounds since they were food system advocates and farmers or academics and consumers. In addition, workshop participants worked with multiple actors in the food system (e.g., producers, distributors, retail food stores) and, therefore, indirectly represented the experiences of multiple food system actors. Most participants worked on food system issues at the local and national levels, but some worked in international settings. Most participants represented organizations/companies/institutions based in North America (mainly the US and some from Canada), but some participants were based in Europe and other parts of the world.

### 2.3. Workshop Organization

The workshop was organized as a series of facilitated discussions that were held over the course of one day (9:30 a.m. to 4:30 p.m. with one hour for lunch). Each facilitated discussion was based on a script. Scripts have been used extensively in group model building workshops [47] and have a specific set of inputs, processes, and outputs. The research purpose of the workshop was the advancement of analyses linking land use, agriculture, food processing, diet, and health for decision-making. The overall aim of the workshop was

to produce a framework that builds and connects ontologies, computable vocabularies, and data sets. We prepared a handout (Supplementary Material) for participants that outlined the draft research question, workshop agenda, and additional resources for the workshop. We disseminated the handout to workshop participants before the start of the workshop.

The draft research question, which was developed by our team before the workshop, was "*Does food composition and nutrient variability in agricultural production impact health in an integrated food system?*" Based on this research question, the workshop began with a concept mapping exercise. The participants identified concepts based on their expertise and knowledge of the foodshed during this activity. The second activity was called "Key Stakeholders with Power vs. Interest Graph". In this activity, the facilitator identified the level of interest and level of power among key stakeholders in addressing the research question. The output of this activity was a set of key stakeholders that were clustered into broader categories (e.g., Business, Labor, Consumers, see Table 1) based on feedback and consensus from participants.

**Table 1.** Broad categories (top row) and mapping of each key stakeholder to each broad category of stakeholders. The broad category titles and set of stakeholder types under each category were developed by the workshop participants; we have listed them here without any modification.

| Business (Industry) * | Government/ Regulators | Healthcare/ Public Health | Business (Farmers) * | Consumers/Activists/ Civil Society | Labor |
|---|---|---|---|---|---|
| Food industry | Lobbyists | Nutritional leaders (pollinators) | Row crop producers, infrastructure invested in corn + soy | Community gardens | Chefs |
| Food brokers | Legislators | Healthcare providers | Farmers | Agriculture researchers | Documented farm workers |
| Transportation sector | Environmental negotiators | Public health departments (CDC) | Plant breeders | Food justice activists | Undocumented farm workers |
| Input providers (e.g., Agriculture, chemical companies) | Government (norms, initiatives) | Health research laboratory | Urban gardeners | Low-income communities | Food system labor (includes farm workers) |
| Retailers | | Healthcare insurance | | WIC recipients (single, urban mothers with young kids) | |
| Media | | Nutritionists | | Indigenous youth (Native Americans) | |
| Catering industry Logistics firms Entrepreneurs | | | | Consumer | |

* The participants delineated between these two categories by identifying in () that Business (industry) corresponded to organizations that include many different types of companies across the food system and Business (farmers) corresponded to organizations that were actively involved in food production as their primary type of business.

The last activity involved multiple steps and was designed to elicit terms that described how power influenced the relationships among these broad categories of stakeholders. The multi-step approach that we took was meant to introduce participants to the concept of ontologies and the PPOD ontology, using non-technical language and in an incremental manner. The *first step* involved asking participants to describe the ways or means by which stakeholders influenced each other. For this step, we limited the stakeholder categories to three very broad groups (business, government, and consumer) because our goal was to slowly introduce the participants to ontologies by first showing how words were used to describe how power was exercised among these three types of stakeholders.

The *second step* was to obtain additional information from participants about the current power relationships among the stakeholder categories. This activity was a modified version of a common script used in group model building workshops called "Connection Circles", where participants write key factors based on a problem around a circle and draw arrows between factors to identify the influence of one factor on another and add a plus or minus to indicate whether the relationship between variables is positive (i.e., increase in one factor leads to an increase in another factor) or negative (i.e., increase in one factor leads to a decrease in another factor). Differently, we prompted participants to get into smaller groups, draw a circle with groups of stakeholders listed in Table 1 (top row) around the circle, and identify which groups had power over each other with a directed arrow. We also asked participants to draw arrows (in a different color) to identify power relationships that would ensure a more equitable distribution of power in the foodshed in some future scenarios. After participants were finished drawing their circles in the smaller group setting, each group was asked to share their answers with all workshop participants. Based on these discussions, the facilitator created a graphic that described current and future power relationships between each category of stakeholders.

The *last step* involved asking participants to individually write out as many statements as possible in the following format: (*stakeholder A*) (*a verb to describe how influence or power is exerted on*) (*stakeholder B*). In ontology development, this is akin to describing the relationship (using a verb) between two objects (using nouns; (*stakeholder A*) and (*stakeholder B*)). We were able to extract verbs to describe how power and influence are exerted among different stakeholders by collecting participant responses. In the future, we plan to use this collection of verbs and nouns to initiate the development of the class Power within the existing PPOD ontology. For the workshop, we simply compiled the verbs and asked participants to describe examples from their disciplines or real-world experiences that captured the power relationship among food system stakeholders.

All the content in the figures and tables that was generated in each workshop activity was finalized after reaching the consensus of all workshop participants. The consensus was reached among all workshop participants when facilitators of each workshop activity visually and verbally confirmed with participants that they were satisfied with the product of each workshop activity.

## 3. Results

The workshop produced a variety of boundary objects in the form of visualizations, notes, and graphs. We digitized these boundary objects for reporting purposes. We did not change any wording in any of the boundary objects shown below, but added clarification to some of the words where necessary.

From the first activity, a concept map emerged showing the relationships among production, composition and variability, and health (Figure 1). Relationships between each concept were indicated by lines, but we did not ask participants about the direction or magnitude of the relationship. Our main objective for generating a concept map was to engage participants from diverse backgrounds and experiences. We also sought to help participants visually understand how the concepts that they were familiar with were related to concepts from other disciplines or domains. Based on the concept map, which we left up on a separate whiteboard for participants to view and refer to throughout the workshop, participants identified 34 different stakeholders with varying levels of power and interest in addressing the research question (Figure 2).

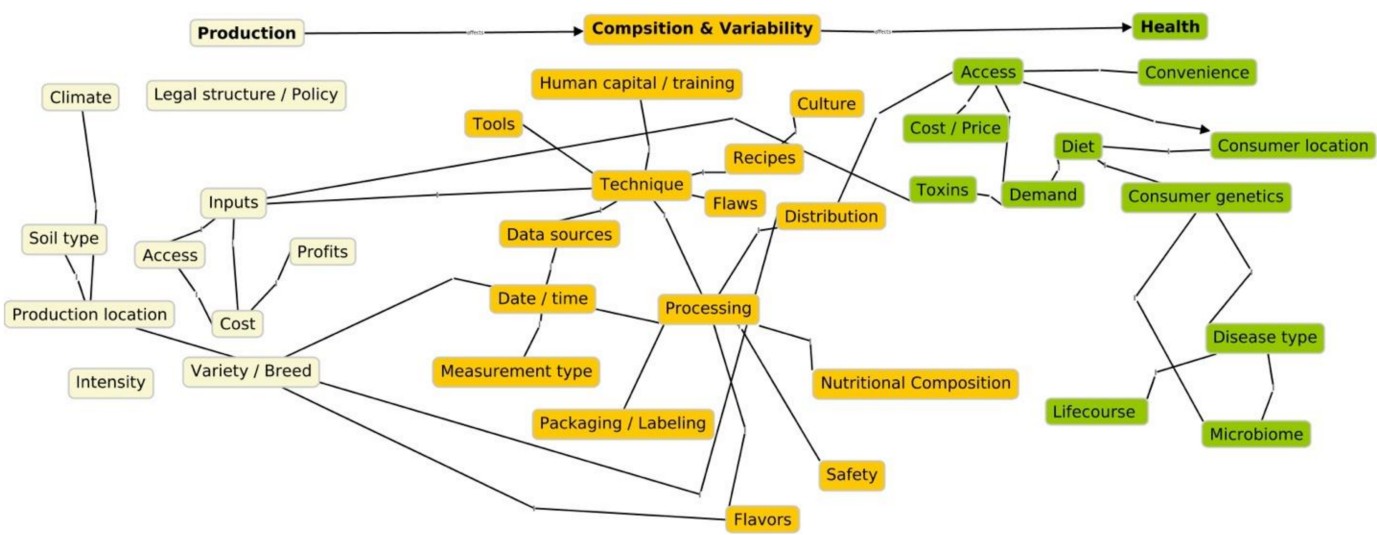

**Figure 1.** Concept map showing the relationship between each concept, which is based on the input received from workshop participants after reviewing the research question for the workshop.

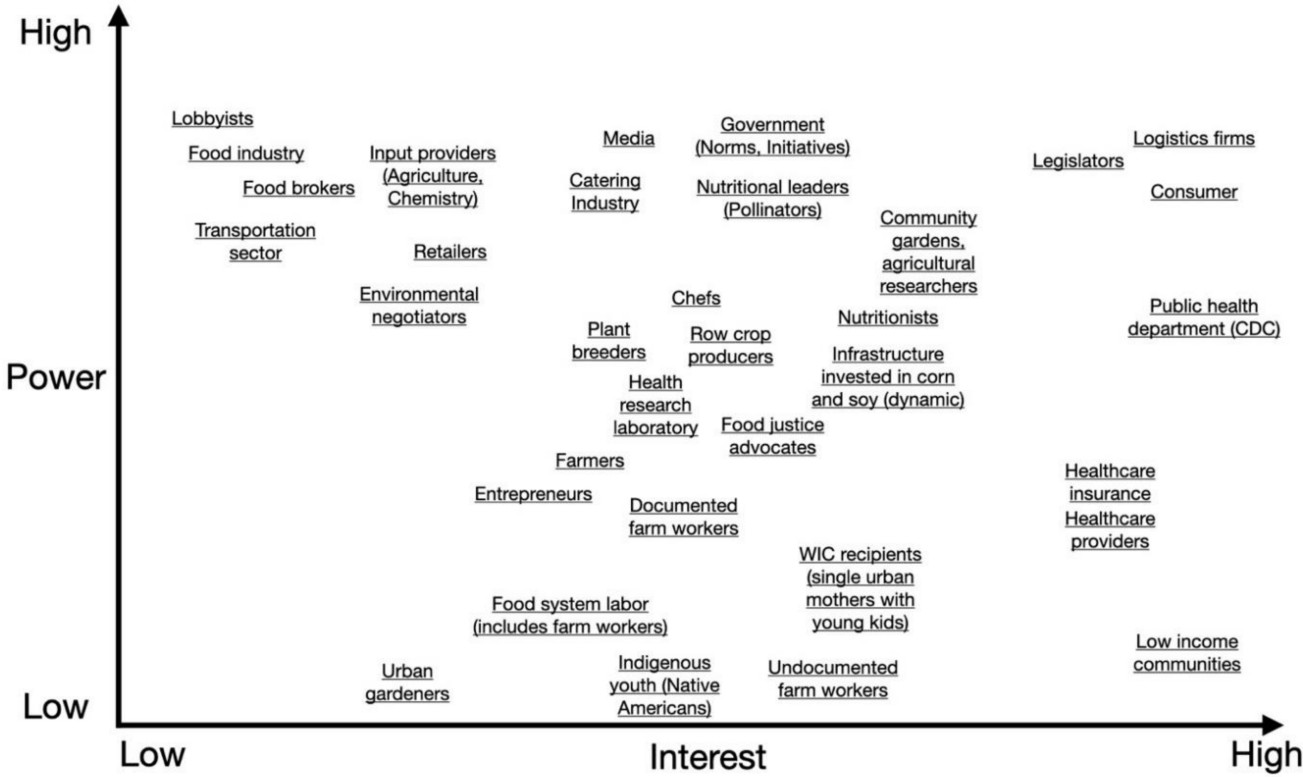

**Figure 2.** Key stakeholders and their level of power and level of interest based on the research question.

The power vs. interest graph was the output from the second activity of the workshop. The graph showed that lobbyists and the food industry had the highest power and lowest interest (top left in Figure 2), low-income communities had the highest interest and lowest level of power (bottom right in Figure 2), and logistic firms, legislators, and consumers had the highest power with the highest interest (top right in Figure 2). Participants grouped the 34 stakeholders that they identified into six broad categories of stakeholder. These stakeholder categories were business (industry), business (farmers), government/regulators,

healthcare/public health, consumers/activists/civil societies, and labor (Table 1, top row). See note in Table 1 for the difference between business (industry) and business (farmers).

One output from the third activity (Figure 3), which further simplified the six stakeholder categories from Table 1, identified feedback loops among business, government, and consumer stakeholders. Specifically, participants identified how consumers influenced businesses through their money and their voice and influenced government through their money, voice, and votes. Voice [48] was broadly meant to convey the social capital of a stakeholder category that the stakeholder may use to influence the actions of another stakeholder. Examples of how consumers used their voice to influence other stakeholders included consumer feedback, online reviews, and advocacy. Participants also highlighted that business stakeholders influenced consumers through products and marketing and influenced government through votes. Examples of votes were lobbying efforts and other ways that businesses influenced government (broadly including agencies, elected and unelected officials, and regulators) activities, decisions, and policies. Lastly, we observed in Figure 3 that the government influenced both businesses and consumers through its regulatory authority. These broad types of influences or expressions of power, such as money, votes, and voice, among stakeholders, were investigated further in the last workshop activity.

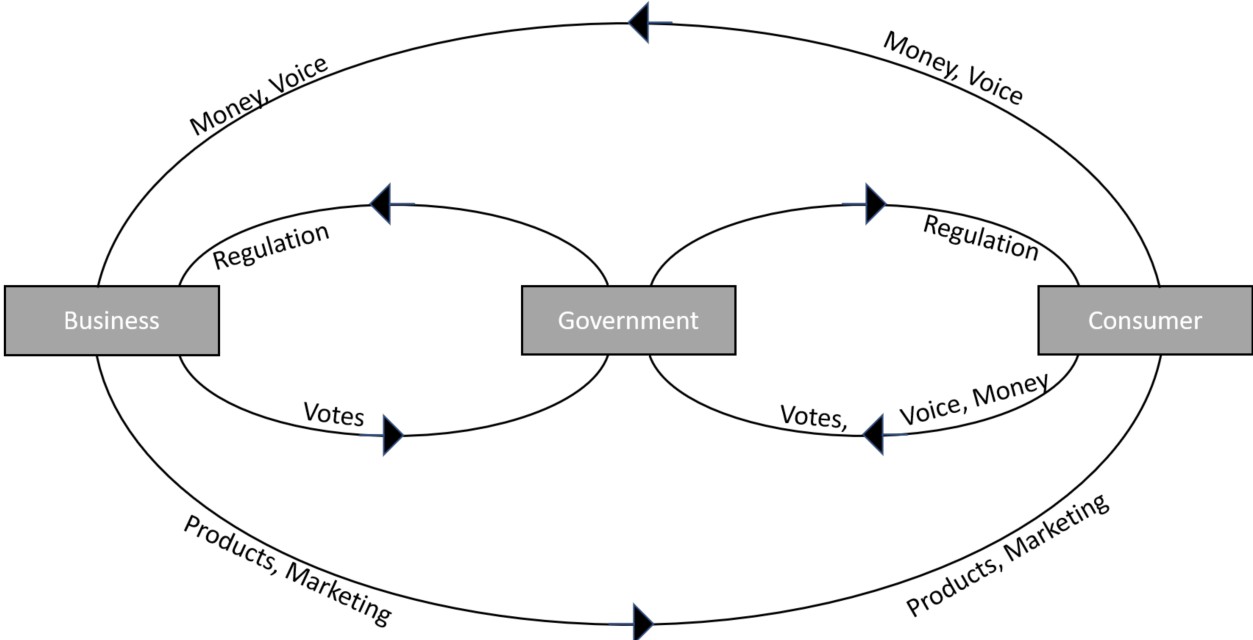

**Figure 3.** Diagram showing how Business, Government, and Consumer stakeholders (grey boxes) influence each other using words to describe the type of the influence. The words in the diagram were based on participant feedback and we have not changed them in the figure. Arrows indicate the direction of influence. For example, "Business" influences "Government" through "votes" (e.g., political lobbying rather than direct votes during an election). "Money" is considered strictly in terms of payment of taxes to the government and for goods and services to business, although it is implicit in marketing, votes, and regulation. "Regulation" is meant to convey policies, incentives, and regulations.

Another output from the third activity identified the current and future influences or power relationships among the six broad stakeholder categories (Figure 4). Through this activity, the diagram of current relationships (black arrows in Figure 4) showed that business (industry) influenced the greatest number of stakeholder groups (e.g., government, farmers, and laborers). Comparatively, the influences of business (farmers), laborers, consumers, and healthcare sectors are currently limited. In terms of future relationships (red arrows in Figure 4) that may lead to a more sustainable and equitable foodshed, we observed that

most participants indicated a need for greater power and influence for consumers, workers, and the health sector vis-a-vis government and business (industry) stakeholders.

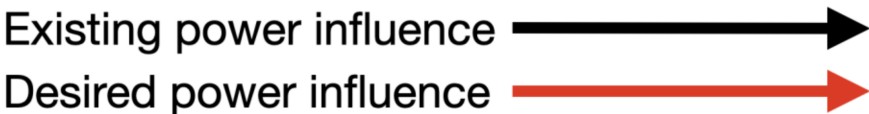

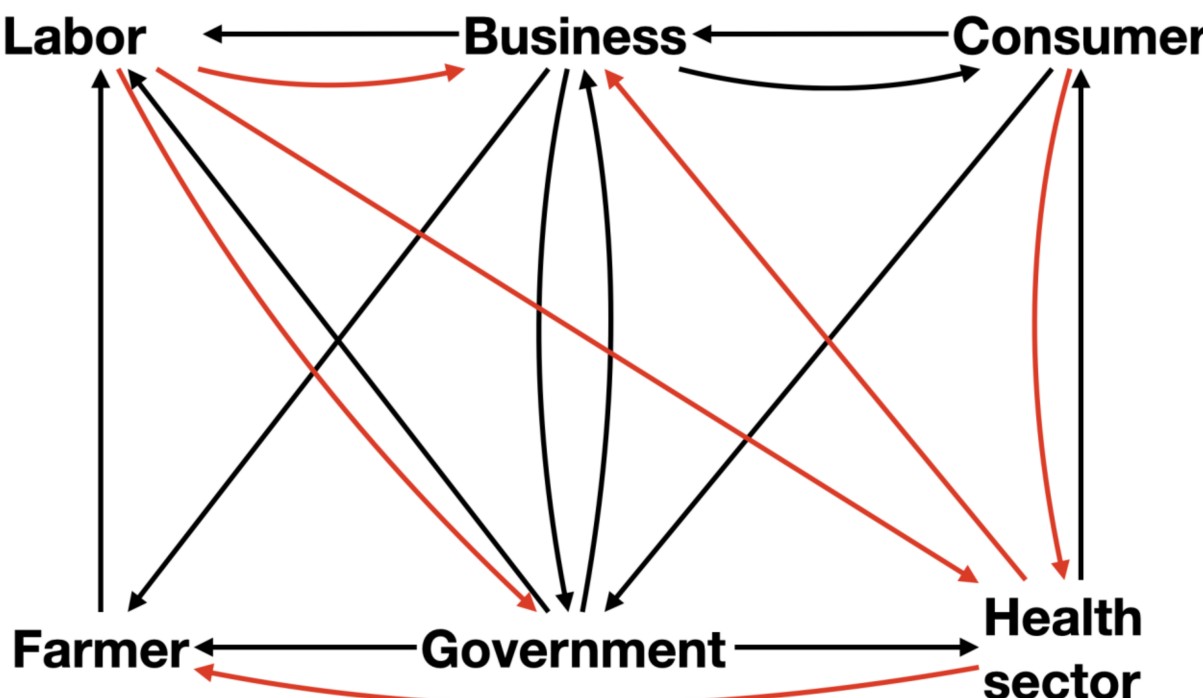

**Figure 4.** Current and desired influences based on power/agency among the broad categories of stakeholders. For example, the "Health sector" may desire to influence the number of pesticides that a "Farmer" uses in food production. While this may be happening in some contexts, the red arrows were meant to also convey a higher level of influence among stakeholders.

The last output from the third activity was the set of statements that further clarified the relationship, in terms of power, agency, and influence, among the six broad stakeholder categories (Figure 5). Based on statements from participants, we noted twenty different statements in the following format: noun → verb → noun, where nouns were objects (e.g., stakeholders) and verbs described the nature of the relationship between two objects in terms of current power relationships. The most common verbs were "funding" and "regulating". Less common verbs were "demonstrating", "advising" and "training". Based on these verbs, the participants generated five broad categories to describe the nature of the power relationships among groups of food system stakeholders: providing the infrastructure; regulation and mandates, media action; providing funding; and withholding resources.

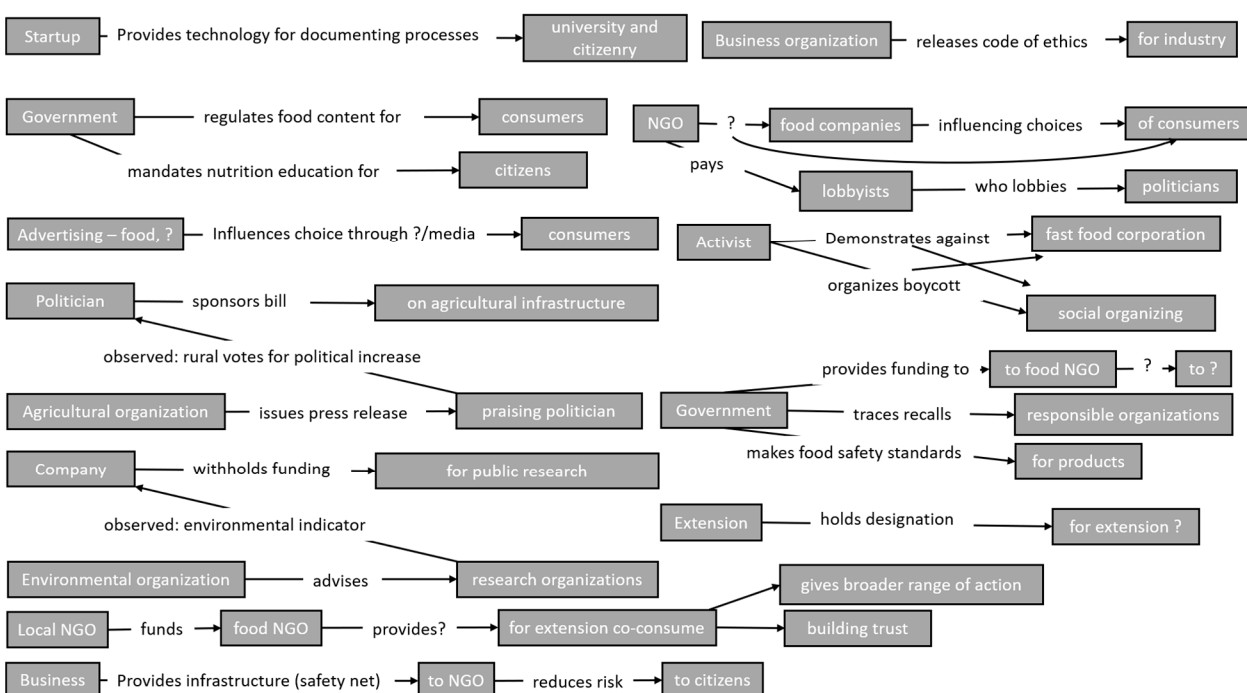

**Figure 5.** Statements produced by participants about how key stakeholder groups influence each other using specific verbs. The collection of verbs will be used in future work to develop the class Power within the PPOD ontology.

## 4. Discussion

In the context of the initial research question, our results highlighted current imbalances in power relationships among food system stakeholders. Although the question was specific to certain components of the food system (e.g., food composition and nutrient availability), the concept map (Figure 1) identified connections among many different parts of the larger food system, and key stakeholders varied in their levels of power and interest (Figure 2). Going one step further, the information that we collected on current relationships among food system stakeholders (Figures 3 and 4) highlighted the importance of increasing the level of power among key stakeholders, such as consumers, workers or labor, and the health sector. Our findings (in Figures 2 and 3) highlighted how those who currently have higher levels of power may be able to reinforce their level of power through the positive feedback loops (Figure 3). Collectively, these findings point toward a path for re-orienting current power relationships (Figure 4) in a manner that could lead to more equitable and sustainable solutions in food systems.

An insight from the workshop was the diverse set of words used by participants to describe how power or influence is practically applied among stakeholders. For example, the statement in Figure 5 that "Government regulates food content for consumers" identifies the role of regulation in terms of how a government entity or stakeholder (e.g., Food and Drug Administration) has direct power and influence over food manufacturers and indirectly on consumers (e.g., people who eat food that is regulated for its quality, safety, and nutritional value by the government). Another example is the statement "Advertising food influences choice through media of consumers". In this statement, participants suggested that it was the choice of consumers that is influenced by food advertisers through any of the several types of media (e.g., news, print, social). The words "choice", "regulate" and words like them, respectively, including other tenses (e.g., choose) or synonyms (e.g., control or police), provide a useful starting point for developing a class Power within the PPOD ontology. Another way to go about creating the shared vocabulary for the class Power would be for developers of PPOD to look up synonyms for the "power" from the dictionary or select concepts from the relevant literature. This alternative approach would have been

much less time-consuming than organizing a workshop. In contrast, we engaged directly with food system stakeholders and empowered them to identify the nouns and verbs by "workshopping" concepts with participants and, thereby, creating a shared vocabulary for the class Power. Our approach started with community engagement rather than the typical research paradigm of generating knowledge and solutions and then going to the community to test their usefulness and utility in real-world settings.

Our current study is an initial attempt to define the class Power within PPOD. We envision a future scenario where a fully developed and instantiated version of the class Power within the PPOD ontology may allow food system researchers and stakeholders alike to elucidate the power relationships between organizations and people. Knowledge about such relationships may be used to understand how power relationships may promote or block the sharing of datasets among food system stakeholders. We plan to use the class Power based on the initial set of verbs gathered through the last workshop activity (Figure 5) and enhance the vocabulary for the class Power (and PPOD) through additional testing with stakeholders through similar workshops, interviews, and focus groups. The following are two examples of how such a scenario could play out in real-world settings.

For the first example, imagine a food system stakeholder (provisionally, call them Organization A) who seeks to answer a question for which they do not have data. Organization A could use a PPOD-based visualization to identify relevant datasets to their question and identify relationships between themselves and other organizations (provisionally, call them the set of organizations B = {$B_1$, $B_2$, . . . , $B_i$}, where $i$ is the total number of organizations that have the data sought by Organization A. Using the class Power in PPOD, the type of influence between organizations A and each $B_i$ could be visualized by Organization A to identify which $B_i$ has power relationships that are more likely to lead to the sharing of data sought by Organization A. To extend this example to food system researchers, the class Power in PPOD may allow them to identify which power relationships may promote or make it more difficult to share data among food system actors. The strength of the PPOD ontology is that, in the future, it may allow for such an analysis to be constrained to specific organizations, datasets, people, and projects.

Another example comes from building upon prior work [49] of some of the authors of the current study. In prior work, these authors linked issues and indicators related to sustainable sourcing using the PPOD ontology. In the future, a fully developed class Power for PPOD may be used to identify the power relationships among organizations around a specific sustainability issue(s) or indicator(s). If the identified power relationships are one-sided or non-existent or weak, then collaborative and trust-building activities may be needed to facilitate data sharing around a specific sustainability issue or indicator. Such activities could be conducted by food system researchers or community stakeholders who specifically focus on a specific sustainability issue or track specific sustainability indicators.

Our study had several limitations. First, we conducted the workshop as part of a conference organized by the International Center for Food Ontology Operability Data and Semantics (IC-FOODS). Therefore, the participants who attended the conference and were self-selected to participate in the workshop are not representative of the full set of stakeholders in the food system. Additionally, our participant recruitment strategy may have biased the set of concepts, stakeholders, description of power relationships, and verbs generated to describe the nature of the power relationship. Although we acknowledge this limitation, it is also worth noting that the purpose of the workshop was to test whether the community-based system dynamics approach could be modified to generate knowledge that would ultimately be used to define the class Power within the PPOD ontology. Therefore, we plan to supplement the initial set of knowledge products/boundary objects generated during the workshop described here with ongoing work that will involve more representative food system stakeholders. Another limitation of our study was the lack of follow-up with participants to validate our interpretation of the boundary objects that they co-created during the workshop. This limitation was mainly due to the nature and format of the workshop, which was a half-day workshop at the end of a multi-day conference.

One consequence of this limitation may be that our interpretation of the verbs identified in Figure 5 may not be reliable because we did not seek further clarification from participants about the reasons behind verbs and nouns they wrote about during the workshop activity. Lastly, a limitation of our study was that the research question was generated by the study team rather than the workshop participants. As a result, it is possible that some participants may not have been very familiar with the issues and topics. This may have been a source of confusion in participant responses during workshop activities, such as the concept mapping activity.

## 5. Conclusions

We designed and implemented a workshop to evaluate the role of power in shaping and solving challenges in a Smart Foodshed. A major takeaway was that the current imbalances in power relationships among food system stakeholders point toward a path for re-orienting current power relationships (Figure 4) in a manner that could lead to sustainable solutions to food systems challenges. Given the limitations of our study, this conclusion is specific to the research question that we used in the workshop. Our methodology (i.e., a workshop based on community-based participatory modeling) was a novel contribution to ontology development, which largely remains an academic exercise rather than one that involves community stakeholders from the beginning of the ontology development process. While our study did not include some food system stakeholders (e.g., farmers and consumers), we plan to include them as we build on this initial application of participatory approaches to ontology development. As a future direction, the class Power could enable adaptive resiliency in a Smart Foodshed. This might be undertaken by cataloging power relationships among foodshed stakeholders in a way that simultaneously reveals something about the interactions among people (who are the stakeholders), projects (that people or organizations carry out), organizations (where people work, projects are implemented and data are generated), and datasets (that are generated, maintained, analyzed, integrated, and used for policy and decisions). Such an advancement in our knowledge via ontologies may have important and novel implications for sustainability science.

The visualization of power relationships, in addition to other types of relationships among food system stakeholders via the PPOD ontology may offer a new way to identify and amplify the voices of those with less power in the foodshed. Such a contribution to the literature of food system sustainability is novel because one of the qualities of a Smart Foodshed is adaptive resiliency. Adaptive resiliency is the ability of a system to adapt to changes in order to remain resilient after a stressor event. Therefore, amplifying the voices of those with less power may prevent the transmission of inequities in the design, governance, operation, and policies of foodsheds and, ultimately, bring about a truly Smart Foodshed.

**Supplementary Materials:** The following supporting information can be downloaded at: https://www.mdpi.com/article/10.3390/su14052642/s1, Document: Handout for the workshop.

**Author Contributions:** Conceptualization, A.H., A.B., A.D.H., C.H., P.R.H., C.M.R., J.F.Q., T.P.T. and M.C.L.; methodology, A.H., A.B., A.D.H., J.F.Q., C.M.R., M.C.L. and C.H.; formal analysis, A.H. and R.S.; resources, A.B., M.C.L., C.M.R. and T.P.T.; writing—original draft preparation, A.H. and R.S.; writing—review and editing, A.H., A.D.H., C.H., P.R.H., C.M.R., T.P.T. and M.C.L.; visualization, A.H., R.S. and C.H.; supervision, A.H.; funding acquisition, A.H., T.P.T., M.C.L. and C.H. All authors have read and agreed to the published version of the manuscript.

**Funding:** The APC funding was provided by corresponding author Ayaz Hyder.

**Institutional Review Board Statement:** Ethical review and approval were waived for this study due to the nature of the workshop, which was part of a public conference, the lack of any data collection about workshop participants except for the registration information that they provided when signing up to attend the workshop, and the use of a group workshop procedure where any recorded information cannot readily identify the subject (directly or indirectly/linked) and any

disclosure of responses outside of the research would NOT reasonably place workshop participants at risk.

**Informed Consent Statement:** Not applicable.

**Data Availability Statement:** Not applicable.

**Acknowledgments:** We thank the participants of the IC-FOODS conference who spent their time and energy with us during the workshop.

**Conflicts of Interest:** The authors declare no conflict of interest. The funders had no role in the design of the study; in the collection, analyses, or interpretation of data; in the writing of the manuscript, or in the decision to publish the results.

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
