# Peer review of "Design and Implementation of a Workshop for Evaluation of the Role of Power in Shaping and Solving Challenges in a Smart Foodshed"

_sustainability, doi:10.3390/su14052642_

Round 1

Reviewer 1 Report

Thank you so much for the opportunity to review this interesting paper. The paper aims to describe the design and implementation of a workshop, perhaps as "A community-based participatory modeling approach". 

Find my comments below:

  1. On Title - Going through the manuscript, I believe this paper aims to 'describe the design and implementation of a workshop for the purpose of evaluating the role of power...: Thus, a community-based participatory modeling approach is the main thrust of this paper. Hence, how about revising the title in a way to reflects the main objective. 
    It will also be better to establish how the workshop can be equated to a community-based participatory modeling approach.
  2. Line 236-240 - How participatory it would be in absence of the main component of the foodshed, i.e., farmers and consumers? (there is some explanation though there under the limitation)
  3. On figures - Are all the figures (Figure 1 through Figure 5) come out through the consensus among the stakeholders? or there is any heterogeneity among the stakeholders in their understanding/perceptions?
  4. Line 436-437 - "and how those power relationships may promote or block the sharing of datasets generated by projects led by participating organizations" Is this really covered in the result section? or explaining this in terms of 'Power" (Figure 2). The one with more power has more say in either promoting or blocking the sharing...? or based on the findings presented in Figure 5?

    It will be interesting to see how "promotion or blocking of datasets" is conceptualized and explain the process "...PPOD ontology will allow... to elucidate the power relationship..., and how those power relationships may promote or block the sharing of datasets..."

  5. I was quite surprised to see so many typographical errors. 
  6. The conclusion could be revised highlighting the major takeaways or the implication amidst the given limitations.

Looking forward.

Author Response

Response to Reviewer 1

  1. On Title - Going through the manuscript, I believe this paper aims to 'describe the design and implementation of a workshop for the purpose of evaluating the role of power...: Thus, a community-based participatory modeling approach is the main thrust of this paper. Hence, how about revising the title in a way to reflects the main objective. 
    It will also be better to establish how the workshop can be equated to a community-based participatory modeling approach.

R1: We thank the reviewer for this suggestion. We have revised the title of the manuscript to the following “Design and Implementation of a Workshop for Evaluation of the Role of Power in Shaping and Solving Challenges in a Smart Foodshed”.

  1. Line 236-240 - How participatory it would be in absence of the main component of the foodshed, i.e., farmers and consumers? (there is some explanation though there under the limitation)

R2: We agree with the reviewer that our workshop did not directly include individuals who identify primarily as farmers, consumers, or consumer advocates. This initial workshop is the start of our project on identifying the role of power in a Smart Foodshed and we plan to hold an additional workshop that is likely to include diverse stakeholders in the food system including farmers and consumers. We have added the following sentences around lines 243-246 to address this comment.

      “Some participants represented multiple backgrounds since they were food system advocates and farmers or academics and consumers. In addition, workshop participants worked with multiple actors in the food system (e.g., producers, distributors, retail food stores) and therefore, indirectly represented the experiences of multiple food system actors.”

  1. On figures - Are all the figures (Figure 1 through Figure 5) come out through the consensus among the stakeholders? or there is any heterogeneity among the stakeholders in their understanding/perceptions?

R3: We appreciate the reviewer’s comment. We can confirm that all figures come through consensus among the stakeholders. We have clarified this on lines 318-322 with the following sentences.

      “All content in the figures and tables that were generated through these workshop activities were based on the consensus of all workshop participants. The consensus was reached among all workshop participants when facilitators of each workshop activity visually and verbally confirmed with participants that they were satisfied with the product of each workshop activity.”

  1. Line 436-437 - "and how those power relationships may promote or block the sharing of datasets generated by projects led by participating organizations" Is this really covered in the result section? or explaining this in terms of 'Power" (Figure 2). The one with more power has more say in either promoting or blocking the sharing...? or based on the findings presented in Figure 5?

      It will be interesting to see how "promotion or blocking of datasets" is conceptualized and explain the process "...PPOD ontology will allow... to elucidate the power relationship..., and how those power relationships may promote or block the sharing of datasets..."

R4: This comment is not covered in the Results section and was written as a discussion of how future and complete versions of PPOD with the class Power may be used in practice. We have revised lines 443-472 to clarify this point in the revised manuscript and added text to describe how we think PPOD ontology might help to elucidate power relationships that promote or block the sharing of datasets.

  1. I was quite surprised to see so many typographical errors. 

R5: We have revised the manuscript o fix the typographical errors.

  1. The conclusion could be revised highlighting the major takeaways or the implication amidst the given limitations.

R6: We have revised the Conclusion section to highlight the major takeaways and implications of our findings considering the limitations of the study.

Reviewer 2 Report

Type of the Paper (Article)

Sustainability-15

Identifying the Role of Power Relationships in Shaping and Solving Challenges in the Foodshed: A Community-Based Participatory Modeling Approach

Power imbalances are ubiquitous across the food system, crossing local, regional, national, and global scales. The collected information on specific relationships between broad categories of stakeholders highlighted needs for initiatives and activities to increase the types and varieties of power especially across consumers, farmers, and labour stakeholder groups. Applying this framework, workshop participants were able to envision new power relationships and a fundamental re-orienting of current power relationships capable of valorizing food system sustainability/resiliency and especially the health of its workers and consumers.

Such a classification system could enable the reimagining of existing power relationships within the food systems to make progress towards greater equity, diversity, transparency, and accountability of food system actors. It is hoped that our work can provide tools to enhance organizing and collective action to confront powerful vested interests standing in opposition to the public good.

Keep the sentences shorter.

English must be revised, avoiding any misleading.

Looks like a workshop report.

Data looks not sufficient.

Author Response

Response to Reviewer 2

Keep the sentences shorter.

R: We have revised the manuscript to address this comment.

English must be revised, avoiding any misleading.

R: We have revised the manuscript to address this comment.

Looks like a workshop report.

R: We have revised the title of the manuscript to reflect that the study was about the design and implementation of a workshop.

Data looks not sufficient.

R: We are not clear on what types of additional data the reviewer would like us to show in a revised manuscript. We have provided all the data that we collected from the workshop in the current version of the manuscript and did not collect any additional data that we could include as part of the revised manuscript. We hope that this is a satisfactory response to the reviewer’s comment.

Round 2

Reviewer 2 Report

The paper has been improved well and can be accepted in the present form.

Author Response

The paper has been improved well and can be accepted in the present form.

R: We thank the reviewer for this encouraging comment.